# Training and Body Composition during Preparation for a 48-Hour Ultra-Marathon Race: A Case Study of a Master Athlete

**DOI:** 10.3390/ijerph16060903

**Published:** 2019-03-13

**Authors:** Pantelis T. Nikolaidis, Celina Knechtle, Rodrigo Ramirez-Campillo, Rodrigo L. Vancini, Thomas Rosemann, Beat Knechtle

**Affiliations:** 1Exercise Physiology Laboratory, 18450 Nikaia, Greece; pademil@hotmail.com; 2Medbase St. Gallen Am Vadianplatz, 9001 St. Gallen, Switzerland; celina.knechtle@medbase.ch; 3Laboratory of Human Performance, Quality of Life and Wellness Research Group, Department of Physical Activity Sciences, Universidad de Los Lagos, Osorno 5290000, Chile; r.ramirez@ulagos.cl; 4Strength and Conditioning Laboratory of the Center of Physical Education and Sport, Federal University of Espírito Santo, Vitória-ES 29075-910, Brazil; rodrigoluizvancini@gmail.com; 5Institute of Primary Care, University of Zurich, 8006 Zurich, Switzerland; thomas.rosemann@usz.ch

**Keywords:** bio-impedance analysis, exercise intensity, master athlete, ultra-endurance

## Abstract

Although the acute effects of ultra-endurance exercise on body composition have been well studied, limited information exists about the chronic adaptations of body composition to ultra-endurance training. The aim of the present study was to examine the day-by-day variation of training and body composition of a master athlete during the preparation for a 48-hour ultra-marathon race. For all training sessions (*n* = 73) before the race, the running distance, duration, and pace were recorded, and body mass, body fat (BF), body water (%), visceral fat, fat-free mass (FFM), four circumferences (i.e., waist, upper arm, thigh and calf), and eight skinfolds (i.e., chest, mid-axilla, triceps, subscapular, abdomen, iliac crest, thigh and calf) were measured accordingly in a 53-year-old experienced ultra-endurance athlete (body mass 80.1 kg, body height 177 cm, body mass index 25.6 kg·m^−2^). The main findings of the present study were that (a) the training plan of the ultra-endurance master athlete followed a periodization pattern with regard to exercise intensity and training volume, which increased over time, (b) the body mass, BF, and FFM decreased largely during the first 30 training sessions, and (c) the circumferences and skinfolds reflected the respective decrease in BF. The findings of this case study provided useful information about the variation of training and body composition during the preparation for an ultra-marathon race in a male master ultra-marathoner. The preparation for an ultra-endurance race seems to induce pronounced changes in body mass and body composition.

## 1. Introduction

Exercise is defined as ultra-endurance when it lasts more than six hours [1]. Consequently, this long duration of exercise results in energy deficit (i.e., negative energy balance) typically expressed by a reduction in body mass, body fat (BF), and fat-free mass (FFM) [2]. It seems that low BF results from the chronic adaptation to ultra-endurance training that might explain its relationship with performance. For instance, race completion in 100 km [3] and 24-hour ultra-marathon running [4] negatively correlates with BF, i.e., the lower the BF, the faster the race time. Therefore, it would be of great interest to estimate the chronic adaptations of body composition to ultra-endurance training.

So far, many studies have examined the acute responses of body composition to ultra-endurance exercise. The decrease in body mass during an ultra-endurance race might even be >5% [5]. Finishing a 24-hour ultra-marathon (122–208 km) [6] and a 230-km run [7] resulted in a decrease in body mass by 1.7% and 1.0–2.5%, respectively, whereas the Antarctic race (2–4 weeks) induced decreases in body mass, BF, and FFM [8]. Furthermore, a reduction in both BF and FFM has been observed in a 5-day run (338 km) [9]. Thus, it was suggested that the exercise-induced decrease in body mass reflected a corresponding decrease in both BF and FFM.

Although the abovementioned studies have enhanced our understanding of the acute responses to ultra-endurance races, limited information (e.g., open water swimming) [10] exists about the chronic adaptations of body composition to ultra-endurance training. It should be highlighted that ultra-marathon runners apply a larger training volume and lower exercise intensity than marathon runners [11]. Considering the increased number of those participating in ultra-endurance races and the concomitant energy deficit in these races [12], it would be of great practical value for sport and exercise science practitioners working in this sport to be aware of the effects of training on body composition in ultra-endurance athletes.

Therefore, the main aim of the present study was to examine the variation in body composition during the preparation period for an ultra-endurance race (48-hour ultra-marathon) of a master athlete, defined as an athlete older than 35 years [13]. A secondary aim was to study the agreement of four (i.e., body mass index—BMI, skinfold thicknesses, circumferences, and bio-impedance analysis) commonly used assessment methods of body composition. It was hypothesized that body mass and BF would decrease during this period due to the increased exercise-induced caloric consumption, and all measures of body composition should record similar trends of changes in these body composition parameters.

## 2. Materials and Methods

### 2.1. Study Design

To examine the variation of training characteristics and body composition during the preparation for a 48-hour ultra-marathon, a longitudinal case study design was applied in which an ultra-endurance athlete was monitored in each training session. The participant was experienced, injury-free and did not receive any medication during the study period. After having been informed about the benefits and risks of this study, the participant provided written informed consent. The study design was in accordance with the Declaration of Helsinski and was approved by the local institutional review board EKOS (Ethikkommission Ostschweiz). All experimental procedures were conducted from October 2017 to January 2018, the month when the race was performed.

### 2.2. Participant

We studied a 53-year-old male ultra-endurance athlete (body mass 80.1 kg, body height 177 cm, body mass index 25.6 kg·m^−2^) who was practicing ultra-endurance training and participating annually in several 6-hour, 12-hour, and 24-hour ultra-marathons. Between 1998 and 2017, he had participated in five 6-hour runs, fourteen 12-hour runs, and six 24-hour runs. He started in these runs before (spring) and after (autumn) his triathlon season as an Ironman triathlete during summer. In contrast to 2017 when he competed in three 24-hour ultra-marathons, his plan was to compete in 2018 in a 48-hour ultra-marathon. Compared to his previous preparations for the 24-hour ultra-marathons, he planned to adopt a similar training program, increasing only the weekly training volume to 110–150 km without changing the running speed during the last three months prior to the race. His training principle was very simple. During winter, it was not possible for him to cycle due to weather conditions (i.e., cold, ice) and, therefore, he was mainly running. Generally, he was traveling in the morning by train to work and then in the early evening running at home (~14 km). This kind of training was normal for his winter training. When the race came nearer, he was running both ways in the morning and the evening and was also performing training sessions on Saturday and Sunday. The length of his training lap during weekends was similar to his running distance to work. Training for this ultra-marathon started after a long holiday of several weeks without training in autumn after his last triathlon. The first sessions were only ~7 km. His pre-race preparation started on Monday 16 October 2017 and lasted until Sunday 21 January 2018 before he went to the race. After Christmas 2017, he was running every day. A Garmin global positioning system smartwatch vívoactive™ (Garmin, Olathe, KS, USA) was used to maintain a pace of 8–9 min·km^−1^ corresponding to 7.0–7.5 km·h^−1^ as a target running speed for the race. To express running speed in % maximum oxygen uptake (%VO_2max_), we can estimate from a case report where intensity in %VO_2max_ was determined during laboratory testing. An ultra-triathlete competing in a Triple Iron ultra-triathlon (i.e., 11.4 km swimming, 540 km cycling, and 126.6 km running) completed 115 km running in 18:22 h:min corresponding to a mean running speed of ~6.3 km·h^−1^ [14]. Heart rate during the run was ~120 bpm, equal to ~45–55 %VO_2max_. For the present case study, running at 8–9 min·km^−1^ could be considered at an intensity of ~55–65 %VO_2max_. The preparation was performed for the 48-hour ultra-marathon race starting on 26 January 2018, in Athens, Greece, in the context of the 13th Festival Athens Ultramarathon, part of the International Association of Ultrarunners (IAU). The race was to take place in a flat area close to the sea.

### 2.3. Equipment and Protocols

All measurements were performed at the same time of the day (9–10 p.m.) to avoid the effect of diurnal variations of anthropometric characteristics and body composition. Measurements were taken about three hours after dinner and after voiding the urinary bladder. Body mass, BF, percent body water, visceral fat, and FFM were measured using the Tanita BC-545 Bioelectrical Impedance Scale (Tanita Corporation of America Inc., Arlington Heights, IL, USA). Body height was measured using a stadiometer (Tanita HR 001 Portable Height Measure, Tanita Europe, Amsterdam, The Netherlands) to the nearest 1 cm. The BMI was calculated as body mass divided by the square of body height (i.e., kg·m^−2^). The thicknesses of eight skinfolds (i.e., chest, midaxillary, triceps, subscapular, abdomen, suprailiac, thigh, and calf) were measured on the right side of the body by a skinfold caliper (GPM-Hautfaltenmessgerät, Siber & Hegner, Zurich, Switzerland) to the nearest 0.2 mm. The circumference of the waist, upper arm, thigh, and calf was measured using a non-elastic measuring (cm) tape (KaWe CE, Kirchner und Welhelm, Germany) to the nearest 0.1 cm. All anthropometric measurements were performed by the same experienced investigator following the protocol of the International Society for the Advancement of Kinanthropometry (ISAK). The reliability of the investigator regarding measuring skinfold thicknesses of ultra-runners under field conditions has already been determined [15].

### 2.4. Statistical Analysis

All statistical analyses were carried out using GraphPad Prism version 7.0 (GraphPad Software, San Diego, CA, USA) and IBM SPSS v.23.0 (SPSS, Chicago, IL, USA). A non-linear (4th grade) regression analysis examined the variation of training and anthropometric characteristics across pre-race preparation and the coefficient of determination (*R*^2^) was calculated to analyze the proportion of the variance in the training characteristics and body composition that was predictable from the training sessions. Pearson correlation coefficient r was used to examine the relationship among the different assessment methods of body composition. The magnitude of the correlations was evaluated as trivial (*r* < 0.10), small (0.10 ≤ *r* < 0.30), moderate (0.30 ≤ *r* < 0.50), large (0.50 ≤ *r* < 0.70), very large (0.70 ≤ *r* < 0.90), and perfect (*r* ≥ 0.90) [16]. The acceptable type I error was set at *p* < 0.05. 

## 3. Results

The running distance, time, and pace per training session are depicted in Figure 1. The running distance ranged from 6.7 km (8th training session, i.e., 46% of the longest training session) to 14.5 km (67th), the training duration from 0:51 h:min (6th, 8th, and 11th, i.e., 44% of the longest training session) to 1:56 h:min (57th), and the pace from 7:03 min:s.km^−1^ (64th) to 8:27 min:s.km^−1^ (57th, i.e., 20% slower than the fastest training session). The first eleven training sessions had a distance of ~7 km and the following had ~14 km with a small variation between the sessions. Correspondingly, the running time was also doubled in the following sessions compared to the first eleven. The running pace reached a “nadir” when the distance shifted from ~7 km to ~14 km and thereafter increased progressively to reach a “peak” at the 50–60th training sessions, and in the last training session decreased again.

Body mass, body fat, and FFM decreased continuously during the first 30 training sessions; then, body fat increased until the end of the preparation, whereas body mass and FFM reached a “peak” close to the 60th training session and, thereafter, decreased (Figure 2). The analysis of circumferences either in absolute values (Figure 3) or in comparison with baseline values (Figure 4) showed similar trends: all four circumference sites decreased progressively in the first 30 training sessions, and thereafter, presented smaller changes. The sum of the eight skinfolds across time presented a similar trend as the circumferences, i.e., a progressive decrease during the first 30 training sessions and smaller changes in the last period before the race (Figure 5). Nevertheless, this trend was not global for all skinfolds, e.g., two skinfolds (chest and subscapular) did not vary by time. The relationship of BF with the other measures ranged from moderate to large, whereas BMI correlated largely to very largely with BF, skinfolds, and circumferences (Table 1). The Σ8 skinfolds correlated largely to very largely with circumferences.

## 4. Discussion

The main findings of the present study were that (a) the training plan of the ultra-endurance master athlete followed a progressive pattern with regard to exercise intensity and volume, (b) his body mass, body fat, and FFM decreased largely during the first 30 training sessions, and (c) the circumferences and skinfolds reflected the respective decrease in body fat.

The analysis of the training during preparation showed that it followed the commonly used training principles (a progressive increase in volume and exercise intensity). Particularly, the training volume (distance and running time) increased dramatically after the 10th training session, from 7 km to 14 km per training session. Thereafter, the volume of each training session was kept constant until the last training session before the competition. The intensity of training (i.e., running pace) increased after the 10th training session, in line with the increase in volume. It should be highlighted that the running pace—at which the sessions after the first 11 were performed—was the variable that undulated, as evidenced by a lower *R*^2^ value and slight sinusoidal impression of the curve. This pattern of increased volume and intensity has been previously observed in other long-duration endurance sports [17]. Thereafter, the intensity was reduced (i.e., the pacing became slower) toward the 60th training session, for a final increase before competition. The maintenance of a high training intensity before competition is common in different sports, and even as a taper strategy [18]. For the analysis of the pattern of variation of the training intensity, it is important to consider that the athlete’s target competition speed was 8–9 min·km^−1^ corresponding to 7.0–7.5 km·h^−1^. Therefore, the training intensity was highly specific (7–8 min·km^−1^) according to the intended intensity during competition, an important aspect, especially for experienced athletes, such as the master athlete of this study [19].

The pre-race preparation of this master ultra-marathoner was not typical for ultra-marathoners. Most ultra-marathon runners have at least one long run of between 20 and 40 km per week depending on the race and that 14 km is a rather short training run. In addition, most ultra-marathoners do strength, high-intensity training and a variation of speed and distances. Our runner prepared differently for such an ultra-marathon by using his daily commute to and from work as training.

With regards to body composition, it was observed that the decrease in body mass during the first training sessions reflected a decrease in both body fat and FFM. The changes in the four circumferences were in agreement with the corresponding variation in body fat. Circumferences had been used previously in 100-km runners [20] and mountain ultra-marathoners (7 days, 350 km) [21]. Not only the circumferences but also the sum of the eight skinfolds varied similarly to body fat. Nevertheless, three skinfold sites (i.e., triceps, chest, and sub-scapular) showed trivial to small correlations with the other measures, and this relationship was in agreement with the variation in the triceps, chest, and sub-scapular skinfolds across training sessions (Figure 5). An interpretation of this observation might be that the changes in body composition were local and depended either on the parts of the human body participating the most in running (i.e., skinfold sites in legs: thigh and calf) or having the largest stores of fat (i.e., abdomen and iliac crest skinfold sites) and body mass. Particularly, the baseline values of the triceps, chest, and sub-scapular skinfold sites (6–9 mm) were relatively lower than the sites with large stores of body fat (e.g., abdomen 17 mm, iliac crest 15 mm). Accordingly, the thickness of the skinfold sites of the legs and abdomen decreased across training sessions, whereas the triceps, chest, and sub-scapular did not. For instance, it has been shown that the mass of the arms is one-third that of the legs and one-eighth that of the trunk [22]. With regard to the agreement among the various assessment methods of body composition, the lack of perfect correlations among them indicated that they provided complementary information. In addition, the regression line presenting the fluctuation of visceral fat across training sessions followed a similar trend (U shape, Figure 2) as that of body fat. Visceral fat has been characterized by its high metabolic activity, in contrast to subcutaneous fat that has a high storage capacity [23]. Furthermore, it has been shown that visceral fat correlated with subcutaneous and overall fat [24]. Thus, it was reasonable to observe that visceral fat reflected changes in overall fat.

A limitation of the present study was that the caloric intake was not quantified; thus, it was not possible to evaluate the energy deficit and its relationship with the changes in body composition. The athlete was traveling in the morning to work and not eating at home at noon. Therefore, it was not possible to quantify exactly the energy intake during the day. Although it was not possible to exactly quantify energy intake during the day, the athlete reported no changes in eating habits during his pre-race preparation. Future studies would need to investigate in a larger sample whether diet during training would have an influence on body composition in recreational male master ultra-marathoners. On the other hand, one of the strengths of the study was the monitoring of the changes of body composition using four assessment methods (i.e., BMI, bio-impedance analysis, skinfolds, and circumferences) on a daily basis. In addition, the findings have both theoretical and practical interest. This athlete was only running 105 km per week. While elite athletes are competing over longer distances and durations, this training volume is easily in line with those performed by recreational marathon runners and certainly under those associated with elite marathon athletes. On the other hand, they offer practical information for the sports medicine team working with ultra-endurance athletes to develop optimal nutritional strategies in order to counterbalance the energy deficit. The training characteristics of the participant were in agreement with those reported in previous studies on finishers in 50 miles (60–96 miles weekly) [25], 107 km run (max 260 km weekly) [26] and 24-hour run (81 km weekly) [27]. Furthermore, the results highlight the need for further research on the chronic adaptations to training in ultra-endurance athletes since most previous studies have focused on acute effects of a race on physiological parameters [27,28,29].

## 5. Conclusions

In summary, the findings of this case study provide useful information about the variation of training and body composition during the preparation for an ultra-marathon race. This process seems to induce important changes in body mass and body composition highlighting the need for optimal nutrition not only during the race but also for the pre-race period due to the large training volume.

## Figures and Tables

**Figure 1 ijerph-16-00903-f001:**
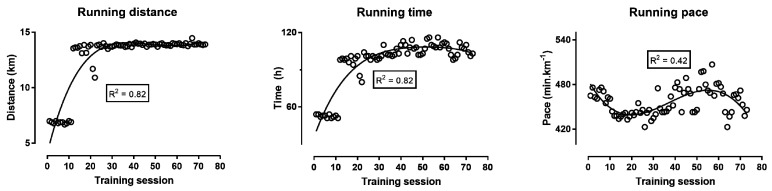
Training characteristics (i.e., running distance, time, and speed per training session) during the preparation for a 48-hour ultra-marathon.

**Figure 2 ijerph-16-00903-f002:**
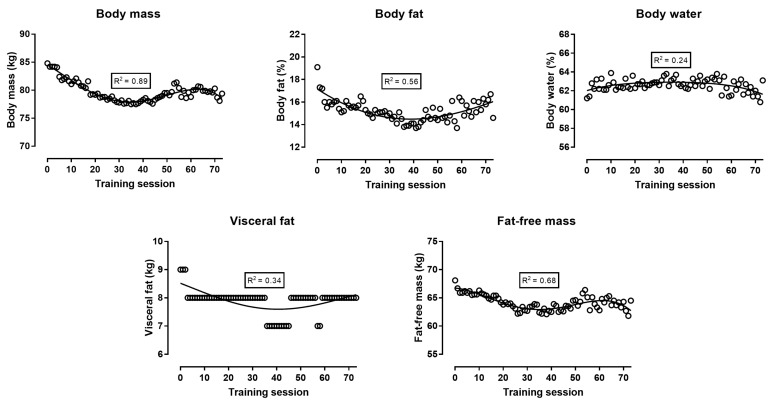
Body composition (i.e., mass, body fat, body water, visceral fat, and fat-free mass) during the preparation for a 48-hour ultra-marathon.

**Figure 3 ijerph-16-00903-f003:**
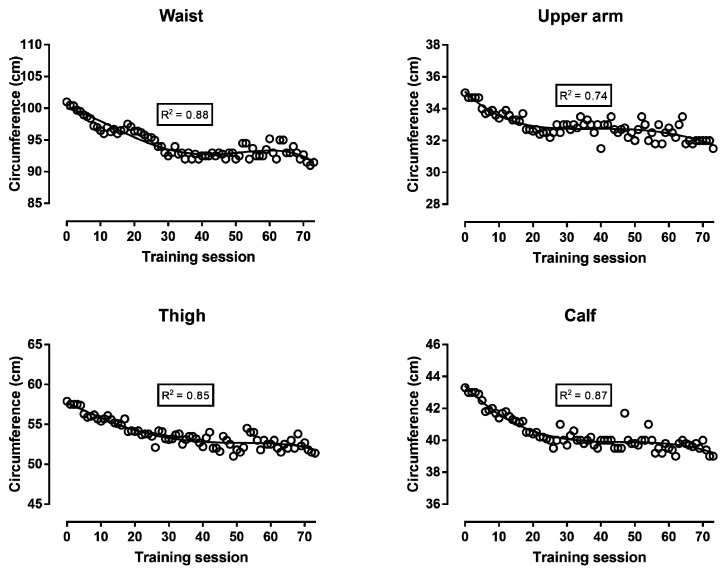
Waist, upper arm, thigh, and calf circumference in absolute values during the preparation for a 48-hour ultra-marathon run.

**Figure 4 ijerph-16-00903-f004:**
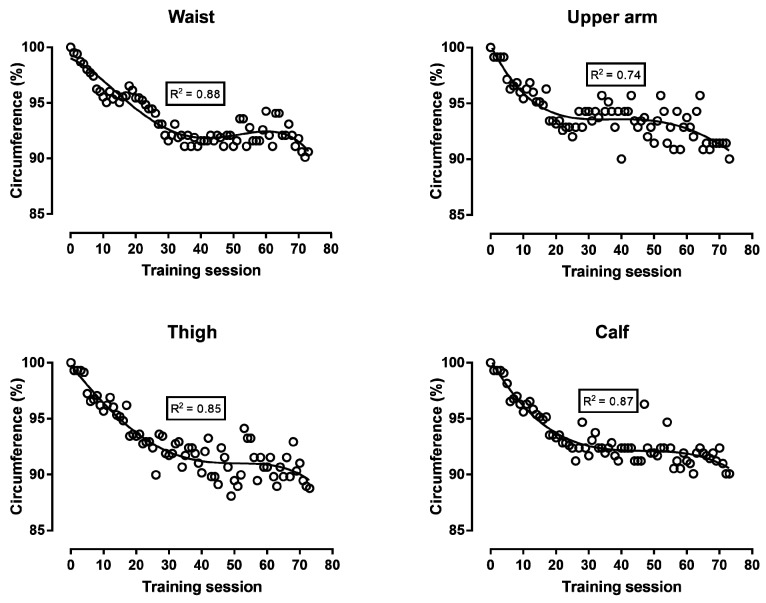
Waist, upper arm, thigh, and calf circumference expressed as a percentage of their baseline values during the preparation for a 48-hour ultra-marathon run.

**Figure 5 ijerph-16-00903-f005:**
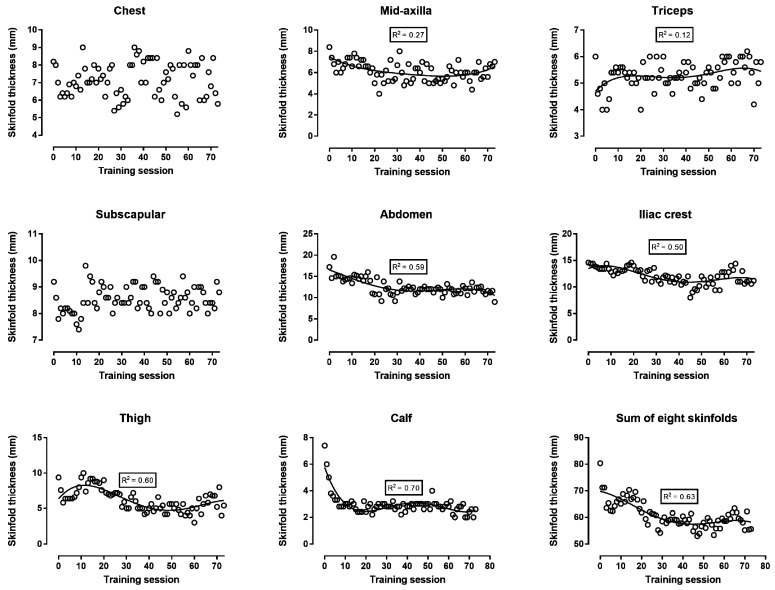
Skinfold thickness in absolute values during the preparation for a 48-hour ultra-marathon run.

**Table 1 ijerph-16-00903-t001:** Correlations (Pearson *r*) among anthropometric characteristics.

Parameter			Skinfolds									Circumferences			
		BMI	Σ8	Chest	Mid-axilla	Triceps	Subscapular	Abdomen	Iliac Crest	Thigh	Calf	Waist	Upper Arm	Thigh	Calf
	BF	0.62 ***	0.59 ***	−0.09	0.34 **	−0.06	−0.04	0.52 ***	0.48 ***	0.41 ***	0.50 ***	0.60 ***	0.36 ***	0.45 ***	0.48 ***
	BMI	−	0.68 ***	−0.13	0.44 ***	−0.15	−0.16	0.71 ***	0.55 ***	0.40 ***	0.57 ***	0.79 ***	0.67 ***	0.76 ***	0.81 ***
Skinfolds	Σ8		−	0.21	0.56 ***	0.04	0.07	0.81 ***	0.70 ***	0.74 ***	0.55 ***	0.77 ***	0.66 ***	0.72 ***	0.69 ***
	Chest			−	−0.07	−0.07	0.19	0.06	−0.07	<0.01	0.10	−0.02	0.05	−0.08	−0.12
	Mid-axilla				−	0.21	−0.03	0.43 ***	0.22	0.30 **	0.33 **	0.33 **	0.41 ***	0.46 ***	0.43 ***
	Triceps					−	0.07	−0.17	−0.06	0.02	−0.12	−0.31 **	−0.30 **	−0.24 *	−0.34 **
	Subscapular						−	−0.03	−0.19	−0.03	−0.02	−0.14	−0.21	−0.20	−0.24 *
	Abdomen							−	0.46 ***	0.44 ***	0.49 ***	0.73 ***	0.72 ***	0.71 ***	0.76 ***
	Iliac crest								−	0.55 ***	0.26 *	0.69 ***	0.50 ***	0.56 ***	0.55 ***
	Thigh									−	0.21	0.57 ***	0.35 **	0.53 ***	0.48 ***
	Calf										−	0.57 ***	0.59 ***	0.55 ***	0.61 ***
Circumferences	Waist											−	0.76 ***	0.85 ***	0.85 ***
	Upper arm												−	0.78 ***	0.82 ***
	Thigh													−	0.91 ***

* *p* < 0.05, ** *p* < 0.01, *** *p* < 0.001, BF = body fat, BMI = body mass index.

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
