# Peer review of "Training and Body Composition during Preparation for a 48-Hour Ultra-Marathon Race: A Case Study of a Master Athlete"

_ijerph, 2019, doi:10.3390/ijerph16060903_

Reviewer 1 Report

Dear authors,

Thank you for having the opportunity to review your manuscript.

Ultramarathon running is popular and research on the training demands and chronic adaptations are still scarce. This case reports highlights this issues and hopefully larger studies will be available soon.

Please see my specific comments below:

Abstract: 

suggest: acute effects… have…

Introduction

Please define age range of master athlete and put into context of typical age range of ultramarathon runners (the age of your runner is in a common age group of ultramarathon)

Equipment and protocols

Were the measurements also taken under the same conditions, meaning e.g. immediately after waking, before going to the toilet or before drinking, breakfast etc? If not please add to limitations.

Results 

I am struggling to understand the training load in km/week and your calculation. You mention in 2.2. that a weekly training load of between 110-150 km/w yet the maximum training distance was 14km/ unit (even if 7 daysX14km=98km). This needs to be clarified or did he have two units/ day? 

Over what time span did he train and how many rest days did he have?

Discussions:

As to my previous comment about training load, you should be more precise about periodization pattern- this at most will be a micro cycle for competition preparation. In my experience this is rather an unusual preparation as most ultramarathon runners have at least one long run (between 20-40km/w, depending on race) and that 14km is a rather short training run. Also most UM runners do strength, high intensity training and variation of speed and distances. I would recommend to touch on this briefly as the reader, not accustomed to UM running, may get an erroneous picture of training demands. Also further on you give references of training loads between 81-260km/w which are in agreement with your participant- please clarify the weekly running distance of your athlete but it doesn’t match the references you cite (21)- line 41ff. Please re-write. 

I would also like to see some more training background information of the athlete, e.g. how long he has been running, what is his usual running distance, how many sessions/w on average (this could be incorporated in the participant section). Did he have a break from running before embarking on this training period, as the anthropometric data seem to suggest that he was deconditioned? 

Unfortunately not having the caloric intake is a major limiting factor of this study on the anthropometric data. Therefore concluding that your case gives practical information about nutritional strategies is not correct. (line 39,40). I agree with the statement but not that this can be concluded from your case study- please re-write. 

Figures and tables are clear and precise. 

Author Response

Open Review

English language and style

( ) Extensive editing of English language and style required
( ) Moderate English changes required
(x) English language and style are fine/minor spell check required
( ) I don't feel qualified to judge about the English language and style

Yes

Can be improved

Must be improved

Not applicable

Does the introduction provide sufficient   background and include all relevant references?

(x)

( )

( )

( )

Is the research design appropriate?

(x)

( )

( )

( )

Are the methods adequately described?

(x)

( )

( )

( )

Are the results clearly presented?

(x)

( )

( )

( )

Are the conclusions supported by the results?

( )

(x)

( )

( )

Comments and Suggestions for Authors

Dear authors,

Thank you for having the opportunity to review your manuscript.

Ultramarathon running is popular and research on the training demands and chronic adaptations are still scarce. This case reports highlights this issues and hopefully larger studies will be available soon.

Please see my specific comments below:

Abstract: 

suggest: acute effects… have…

Answer: We agree with the expert reviewer and changed as suggested

Introduction

Please define age range of master athlete and put into context of typical age range of ultramarathon runners (the age of your runner is in a common age group of ultramarathon)

Answer: We agree with the expert reviewer and added ‘…defined as an athlete older than 35 years.  There is no typical age range, the oldest ultra-marathoners can be 95 years old, see https://www.ncbi.nlm.nih.gov/pubmed/30687109

Equipment and protocols

Were the measurements also taken under the same conditions, meaning e.g. immediately after waking, before going to the toilet or before drinking, breakfast etc? If not please add to limitations.

Answer: We agree with the expert reviewer and added ‘All measurements were performed at the same time of the day (9-10 p.m.) to avoid the effect of the diurnal variation of anthropometric characteristics and body composition. Measurements were taken about 3 hours after dinner and after voiding the urinary bladder’

Results 

I am struggling to understand the training load in km/week and your calculation. You mention in 2.2. that a weekly training load of between 110-150 km/w yet the maximum training distance was 14km/ unit (even if 7 daysX14km=98km). This needs to be clarified or did he have two units/ day? 

Answer: We agree with the expert reviewer and clarify by adding ‘Generally, he was travelling in the morning by train to work and then in the early evening running at home. When the race came nearer, he was running both ways in the morning and in the evening and also performing training units on Saturday and Sunday’.

Over what time span did he train and how many rest days did he have?

Answer: We agree with the expert reviewer and added ‘Generally, he was travelling in the morning by train to work and then in the early evening running at home. When the race came nearer, he was running both ways in the morning and in the evening and was also performing training units on Saturday and Sunday. His pre-race preparation started on Monday 16th October 2017 and lasted until Sunday 21st January 2018 before he went to the race. After Christmas 2017, he was running every day‘.

Discussions:

As to my previous comment about training load, you should be more precise about periodization pattern- this at most will be a micro cycle for competition preparation. In my experience this is rather an unusual preparation as most ultramarathon runners have at least one long run (between 20-40km/w, depending on race) and that 14km is a rather short training run. Also most UM runners do strength, high intensity training and variation of speed and distances. I would recommend touching on this briefly as the reader, not accustomed to UM running, may get an erroneous picture of training demands. Also further on you give references of training loads between 81-260km/w which are in agreement with your participant- please clarify the weekly running distance of your athlete but it doesn’t match the references you cite (21)- line 41ff. Please re-write. 

Answer: We agree with the expert reviewer and added in the discussion ‘The pre-race preparation of this master ultra-marathoner was not typical for ultra-marathoners. Most ultramarathon runners have at least one long run between 20-40 km per week depending on the race and that 14 km is a rather short training run. Also most ultra-marathoners do strength, high intensity training and a variation of speed and distances. Our runner prepared differently for such an ultra-marathon with using his daily way to work as training first in the evening to run back home and later also in the morning to run to work’.

I would also like to see some more training background information of the athlete, e.g. how long he has been running, what is his usual running distance, how many sessions/w on average (this could be incorporated in the participant section). Did he have a break from running before embarking on this training period, as the anthropometric data seem to suggest that he was deconditioned? 

Answer: We agree with the expert reviewer and added in that section ‘Training for this ultra-marathon started after a longer holiday of several weeks without training’. We also added ‘Between 1998 and 2017, he had participated in five 6-hour runs, 14 12-hour runs and six 24-hour runs. He started in these runs before and after his triathlon season as Ironman triathlete during summer’ for his competition background. We also added ‘During winter, it was not possible for him to cycle due to weather conditions and he therefore was mainly running. Generally, he was travelling in the morning by train to work and then in the early evening running at home. This kind of training was normal for his winter training. When the race came nearer, he was running both ways in the morning and in the evening and was also performing training units on Saturday and Sunday. His pre-race preparation started on Monday 16th October 2017 and lasted until Sunday 21st January 2018 before he went to the race. After Christmas 2017, he was running every day. Training for this ultra-marathon started after a longer holiday of several weeks without training’ for the training background.

Unfortunately not having the caloric intake is a major limiting factor of this study on the anthropometric data. Therefore concluding that your case gives practical information about nutritional strategies is not correct. (line 39,40). I agree with the statement but not that this can be concluded from your case study- please re-write. 

Answer: We agree with the expert reviewer and added in the discussion ‘The athlete was travelling in the morning to work and not eating at home at noon. So it was not possible to exactly quantify energy intake during the day’ in the limitations. The sentence in the abstract was shortened.

Figures and tables are clear and precise. 

Answer: We thank the expert reviewer for his/her comment, no changes are required.

Reviewer 2 Report

The authors present a case study of the changes in body composition suffered by a master athlete who is preparing for a 48-hour ultra marathon. Although the data may be interesting to know what happens in the body composition during a training period, the non-presentation of nutritional data is a very great limitation. The authors indicate it in the limitations section. However, its indication in this section does not exempt authors from responsibility since the non-publication of these data makes it impossible for us to understand the results.

On the other hand, the use of BIA and anthropometry is another great limitation. The BIA is dependent on the degree of hydration of the athlete and unless it was done with a methodical protocol the values obtained are difficult to assess. On the other hand, anthropometry is dependent on the technique used and the technical error of measurement that the anthropometrist has. The authors do not specify what protocols they followed to obtain such data.

On the other hand, only the rhythm is indicated as a detail of the training. To know if this rhythm was at 50-60-70-80-90-100% of VO2 max would be of great importance to be able to understand some results.

Finally, I have not been able to understand the R2 that appears in the section of statistical analysis.

Author Response

Open Review

English language and style

( ) Extensive editing of English language and style required
( ) Moderate English changes required
( ) English language and style are fine/minor spell check required
(x) I don't feel qualified to judge about the English language and style

Yes

Can be improved

Must be improved

Not applicable

Does the introduction provide sufficient   background and include all relevant references?

( )

(x)

( )

( )

Is the research design appropriate?

( )

( )

(x)

( )

Are the methods adequately described?

( )

( )

(x)

( )

Are the results clearly presented?

( )

(x)

( )

( )

Are the conclusions supported by the results?

( )

(x)

( )

( )

Comments and Suggestions for Authors

The authors present a case study of the changes in body composition suffered by a master athlete who is preparing for a 48-hour ultra marathon. Although the data may be interesting to know what happens in the body composition during a training period, the non-presentation of nutritional data is a very great limitation. The authors indicate it in the limitations section. However, its indication in this section does not exempt authors from responsibility since the non-publication of these data makes it impossible for us to understand the results.

Answer: We agree with the expert reviewer and added ‘The athlete was travelling in the morning to work and not eating at home at noon. So it was not possible to exactly quantify energy intake during the day’ in the limitations.

On the other hand, the use of BIA and anthropometry is another great limitation. The BIA is dependent on the degree of hydration of the athlete and unless it was done with a methodical protocol the values obtained are difficult to assess. On the other hand, anthropometry is dependent on the technique used and the technical error of measurement that the anthropometrist has. The authors do not specify what protocols they followed to obtain such data.

Answer: We agree with the expert reviewer and explain in the method section ‘All measurements were performed at the same time of the day (9-10 p.m.) to avoid the effect of the diurnal variation of anthropometric characteristics and body composition. Measurements were taken about 3 hours after dinner and after voiding the urinary bladder’ and ‘All anthropometric measurements were performed by the same experienced investigator’.

On the other hand, only the rhythm is indicated as a detail of the training. To know if this rhythm was at 50-60-70-80-90-100% of VO2 max would be of great importance to be able to understand some results.

Answer: We agree with the expert reviewer. For an ultra-marathoner, percent of VO2max is not important. For an ultra-marathoner, it is important to run at a steady pace during the race. This has been shown in many studies investigating pacing of ultra-marathoners.

Finally, I have not been able to understand the R2 that appears in the section of statistical analysis.

Answer: We agree with the expert reviewer and added the explanation in the methods section (“…to analyze the proportion of the variance in the training characteristics and body composition that was predictable from the training units”).

Round  2

Reviewer 2 Report

The authors have done a great job, but they have not answered some of the questions that were posed to them as the ETM of the anthropometrist and if they followed, for example, the ISAK system or another that could be comparable.

On the other hand, the authors indicate that the ultramarathon% VO2 is not important and I disagree with it since depending on which% VO2 train the energy expenditure is greater or not.

Finally, I consider that the limitation of not having the data of the diet is unquestionable and the data of a single athlete loses value. Another thing would have been to present data from many athletes and somehow this could make us see what is the behavior of body composition over a period, but being a case study without nutrition do not bring the results.

Author Response

Open Review

English language and style

( ) Extensive editing of English language and style required
( ) Moderate English changes required
( ) English language and style are fine/minor spell check required
(x) I don't feel qualified to judge about the English language and style

Yes

Can be improved

Must be improved

Not applicable

Does the introduction provide sufficient   background and include all relevant references?

( )

(x)

( )

( )

Is the research design appropriate?

( )

( )

(x)

( )

Are the methods adequately described?

( )

( )

(x)

( )

Are the results clearly presented?

( )

( )

(x)

( )

Are the conclusions supported by the results?

( )

( )

(x)

( )

Comments and Suggestions for Authors

The authors have done a great job, but they have not answered some of the questions that were posed to them as the ETM of the anthropometrist and if they followed, for example, the ISAK system or another that could be comparable.

Answer: We agree with the expert reviewer and added in the method section ‘All anthropometric measurements were performed by the same experienced investigator following the protocol of the International Society for the Advancement of Kinanthropometry (ISAK). The reliability of the investigator regarding measuring skin-fold thicknesses of ultra-runners under field conditions has already determined (Percept Mot Skills. 2010 Aug;111(1):105-6.).

On the other hand, the authors indicate that the ultramarathon% VO2 is not important and I disagree with it since depending on which% VO2 train the energy expenditure is greater or not.

Answer: We agree with the expert reviewer and added in the method section ‘To express running speed in % maximum oxygen uptake (%VO2max), we can estimate from a case report where intensity in %VO2max was determined during laboratory testing. An ultra-triathlete completed 115 km running in 18:22 h:min corresponding to an mean running speed of ~6.3 km/h (Schweizerische Zeitschrift für Sportmedizin und Sporttraumatologie 51:180-187, 2003). Heart rate during the run was ~120 bpm, equal to ~45-55 %VO2max. For the present case, running at 8-9 min·km-1 could be considered as ~55-65 %VO2max’

Finally, I consider that the limitation of not having the data of the diet is unquestionable and the data of a single athlete loses value. Another thing would have been to present data from many athletes and somehow this could make us see what is the behavior of body composition over a period, but being a case study without nutrition do not bring the results.

Answer: We agree with the expert reviewer and added in the limitations ‘Future studies would need to investigate in a larger sample whether diet during training would have an influence on body composition in recreational male master ultra-marathoners’.

Reviewer 3 Report

General comments

This paper is well written but some grammar and word choice changes are required. The researchers have employed a sound longitudinal methodology in tracking body composition changes in response to preparation for an ultra-endurance event, and so sits in complement to the group’s other paper published in Sports which documents the race in question.

Data presentation is of a high standard but please check line spacing of sections as this appears to alter throughout the manuscript.

I have provided line by line comments below, on what was an interesting paper to review.

Line by line comments

Line 21: ‘Training units’ may better be described as training sessions, especially if no cross training or strength training is included in the programme/manuscript.

Line 26: Can I ask why body mass index (BMI) is reported as opposed to body mass in this instance?

Line 28: ‘across time’ may read clearer as ‘over time’

Line 33: You describe the anthropometric changes experienced by the athlete as important and do so again in the conclusion (Line 221) but as you’ve not assessed performance in this manuscript for what reason are they important? A more appropriate word might be ‘pronounced’ i.e. ‘pronounced changes in body mass and body composition…’

Line 38: I would recommend that the ‘the’ at the end of this line be replaced by an ‘a’, and an adverb such as typically also be added, to read ‘…typically expressed by a reduction in body mass…’. I suggest this as not all studies assess all variables, and not every athlete will respond to energy deficits induced by these events in the same way, especially given the chronic adaptations which you go on to describe.

Line 41-42: ‘was related to BF i.e….’ might read better if described more simply as ‘For instance, race completion time in 100km and 24-hour ultra-marathon running negatively correlated with BF.’

Line 48: ‘induced decrease’ should be amended to ‘induced decreases’

Line 52: ‘limited information existed’ should be amended to ‘limited information exists’

Line 57: Do you feel this paper is only suited to strength and conditioning practitioners, or to other professions that support athletes too? Sport and exercise science practitioners may be a broader term, if you feel it appropriate.

Line 73-74: I would recommend amending ‘the participant completed and signed a written informed consent.’ to ‘the participant provided written informed consent.’

Line 82: ‘as an Ironman triathlete during summer.’

Line 91 onwards: please amend ‘training units’ to ‘training sessions’

Lines 123-124: a reference is required to support your inference thresholds for correlation coefficients. I believe it is: Hopkins, W.G.; Marshall, S.W.; Batterham, A.M.; Hanin, J. Progressive Statistics for Studies in Sports Medicine and Exercise Science. Med. Sci. Sport Exerc. 2009, 41, 3–13.

Lines 132-133: the words nadir and peak may be more appropriate versions of ‘bottom’ and ‘top’, respectively.

Figure 2, visceral fat, appears largely stable, but with an area of exception – can this be explained?

Figure 5, chest and subscap appear somewhat random in comparison – is this discussed, can this be explained?

Table 1: Work with editors to agree on appropriate formatting to ensure all data is readable and best presented.

Line 167: I would argue that the training plan is progressive as opposed to periodized, as distance run and session time are relatively stable after the first 11 sessions, but the running pace at which these sessions are performed is the variable that undulates, as evidenced by a lower R2 value and slight sinusoidal impression of the curve

Line 177: following on from Line 176 this currently reads ‘Thereafter, the intensity was reduced increased…’ please delete as appropriate

Line 188: ‘daily way to work…’ may read better as ‘by using his daily commute to and from work as training’. My understanding is that this is common in recreational athletes, but your manuscript may be the first to document this, at least in an ultra-endurance athlete.

Line 193: please add an ‘and’ in between the two types of runners mentioned

Lines 195-198: I would like some more detail here to support your supposition(s). How does your athlete compare to normative data in ultra or endurance runners for instance? If you express these as a percentage change, how do they look? Also, you do not discuss the subscapular for which a correlation coefficient is not reported (neither for chest), but this also shows a great deal of variation. The chest, mid-axilla and tricep appear lean in this individual, which discredits the interpretation of them having the largest stores of fat mass. I think because they are lean, and change in these variables will be expressed as a relatively larger change and so inflates estimation of error within the correlation coefficient calculation.

Line 199: Please change ‘was one third’ to ‘is one third’

Line 205-206: Consider linking these two sentences together ‘Although it was not possible to exactly quantify energy intake during the day, the athlete reported no changes in eating habits during his pre-race preparation.’

Lines 209-211: I would query this comment as your athlete is only running 105km per week. Whilst they are competing over longer distances and durations, this training volume is easily in line with those performed by recreational marathon runners and certainly under those associated with elite marathon athletes.

Author Response

Open Review

English language and style

( ) Extensive editing of English language and style required
(x) Moderate English changes required
( ) English language and style are fine/minor spell check required
( ) I don't feel qualified to judge about the English language and style

Yes

Can be improved

Must be improved

Not applicable

Does the introduction provide sufficient   background and include all relevant references?

( )

(x)

( )

( )

Is the research design appropriate?

(x)

( )

( )

( )

Are the methods adequately described?

(x)

( )

( )

( )

Are the results clearly presented?

(x)

( )

( )

( )

Are the conclusions supported by the results?

( )

( )

(x)

( )

Comments and Suggestions for Authors

General comments

This paper is well written but some grammar and word choice changes are required. The researchers have employed a sound longitudinal methodology in tracking body composition changes in response to preparation for an ultra-endurance event, and so sits in complement to the group’s other paper published in Sports which documents the race in question.

Answer: We agree with the expert reviewer and checked the English spelling again.

Data presentation is of a high standard but please check line spacing of sections as this appears to alter throughout the manuscript.

Answer: We agree with the expert reviewer and managed to solve this problem

I have provided line by line comments below, on what was an interesting paper to review.

Answer: We thank the expert reviewer for his/her comments and worked on each comment.

Line by line comments

Line 21: ‘Training units’ may better be described as training sessions, especially if no cross training or strength training is included in the programme/manuscript.

Answer: We agree with the expert reviewer and changed as suggested.

Line 26: Can I ask why body mass index (BMI) is reported as opposed to body mass in this instance?

Answer: We present body mass, body height and calculate body mass index to show that a master ultra-marathoner can be overweight.

Line 28: ‘across time’ may read clearer as ‘over time’

Answer: We agree with the expert reviewer and changed as suggested.

Line 33: You describe the anthropometric changes experienced by the athlete as important and do so again in the conclusion (Line 221) but as you’ve not assessed performance in this manuscript for what reason are they important? A more appropriate word might be ‘pronounced’ i.e. ‘pronounced changes in body mass and body composition…’

Answer: We agree with the expert reviewer and changed as suggested.

Line 38: I would recommend that the ‘the’ at the end of this line be replaced by an ‘a’, and an adverb such as typically also be added, to read ‘…typically expressed by a reduction in body mass…’. I suggest this as not all studies assess all variables, and not every athlete will respond to energy deficits induced by these events in the same way, especially given the chronic adaptations which you go on to describe.

Answer: We agree with the expert reviewer and changed as suggested.

Line 41-42: ‘was related to BF i.e….’ might read better if described more simply as ‘For instance, race completion time in 100km and 24-hour ultra-marathon running negatively correlated with BF.’

Answer: We agree with the expert reviewer and changed as suggested.

Line 48: ‘induced decrease’ should be amended to ‘induced decreases’

Answer: We agree with the expert reviewer and changed as suggested.

Line 52: ‘limited information existed’ should be amended to ‘limited information exists’

Answer: We agree with the expert reviewer and changed as suggested.

Line 57: Do you feel this paper is only suited to strength and conditioning practitioners, or to other professions that support athletes too? Sport and exercise science practitioners may be a broader term, if you feel it appropriate.

Answer: We agree with the expert reviewer and changed as suggested.

Line 73-74: I would recommend amending ‘the participant completed and signed a written informed consent.’ to ‘the participant provided written informed consent.’

Answer: We agree with the expert reviewer and changed as suggested.

Line 82: ‘as an Ironman triathlete during summer.’

Answer: We agree with the expert reviewer and changed as suggested.

Line 91 onwards: please amend ‘training units’ to ‘training sessions’

Answer: We agree with the expert reviewer and changed as suggested.

Lines 123-124: a reference is required to support your inference thresholds for correlation coefficients. I believe it is: Hopkins, W.G.; Marshall, S.W.; Batterham, A.M.; Hanin, J. Progressive Statistics for Studies in Sports Medicine and Exercise Science. Med. Sci. Sport Exerc. 2009, 41, 3–13.

Answer: We agree with the expert reviewer and added the suggested reference.

Lines 132-133: the words nadir and peak may be more appropriate versions of ‘bottom’ and ‘top’, respectively.

Answer: We agree with the expert reviewer and changed as suggested.

Figure 2, visceral fat, appears largely stable, but with an area of exception – can this be explained?

Answer: We agree with the expert reviewer and addressed this aspect in discussion (“In addition, the regression line presenting the fluctuation of visceral fat across training sessions followed a similar trend (U shape, Figure 2) as that of body fat. Visceral fat has been characterized by its high metabolic activity, in contrast to subcutaneous fat that had high storage capacity [24]. Furthermore, it has been shown that visceral fat correlated with subcutaneous and overall fat [25]. Thus, it was reasonable to observe that visceral fat reflected changes in overall fat.”).

Figure 5, chest and subscap appear somewhat random in comparison – is this discussed, can this be explained?

Answer: We agree with the expert reviewer and discussed further this aspect in the discussion (“and this relationship was in agreement with the variation of triceps, chest and sub-scapular skinfolds across training sessions (Figure 5)” and “Accordingly, the thickness of skinfold sites of legs and abdomen decreased across training sessions, whereas triceps, chest and sub-scapular did not.”

Table 1: Work with editors to agree on appropriate formatting to ensure all data is readable and best presented.

Answer: We agree with the expert reviewer and revised this table in order to be more reader-friendly.

Line 167: I would argue that the training plan is progressive as opposed to periodized, as distance run and session time are relatively stable after the first 11 sessions, but the running pace at which these sessions are performed is the variable that undulates, as evidenced by a lower R2 value and slight sinusoidal impression of the curve

Answer: We agree with the expert reviewer and revised it accordingly in the first and second paragraph of the discussion (“It should be highlighted that the running pace - at which the sessions after the first 11 were performed - was the variable that undulated, as evidenced by a lower R2 value and slight sinusoidal impression of the curve.“).

Line 177: following on from Line 176 this currently reads ‘Thereafter, the intensity was reduced increased…’ please delete as appropriate

Answer: We agree with the expert reviewer and corrected it (i.e. reduced).

Line 188: ‘daily way to work…’ may read better as ‘by using his daily commute to and from work as training’. My understanding is that this is common in recreational athletes, but your manuscript may be the first to document this, at least in an ultra-endurance athlete.

Answer: We agree with the expert reviewer and changed as suggested.

Line 193: please add an ‘and’ in between the two types of runners mentioned

Answer: We agree with the expert reviewer and changed as suggested.

Lines 195-198: I would like some more detail here to support your supposition(s). How does your athlete compare to normative data in ultra or endurance runners for instance? If you express these as a percentage change, how do they look? Also, you do not discuss the subscapular for which a correlation coefficient is not reported (neither for chest), but this also shows a great deal of variation. The chest, mid-axilla and tricep appear lean in this individual, which discredits the interpretation of them having the largest stores of fat mass. I think because they are lean, and change in these variables will be expressed as a relatively larger change and so inflates estimation of error within the correlation coefficient calculation.

Answer: We agree with the expert reviewer and revised this part accordingly. Our intention was to highlight exactly the opinion of the expert reviewer: since chest, mid-axilla and triceps were low stores of fat mass, the training did not reduce them (“Particularly, the baseline values of triceps, chest and sub-scapular skinfold sites (6-9mm) were relatively lower than sites with large store of body fat (e.g. abdomen 17mm, iliac crest 15mm).”). Furthermore, we developed this aspect in the specific paragraph.

Line 199: Please change ‘was one third’ to ‘is one third’

Answer: We agree with the expert reviewer and changed as suggested.

Line 205-206: Consider linking these two sentences together ‘Although it was not possible to exactly quantify energy intake during the day, the athlete reported no changes in eating habits during his pre-race preparation.’

Answer: We agree with the expert reviewer and changed as suggested.

Lines 209-211: I would query this comment as your athlete is only running 105km per week. Whilst they are competing over longer distances and durations, this training volume is easily in line with those performed by recreational marathon runners and certainly under those associated with elite marathon athletes.

Answer: We agree with the expert reviewer and changed as suggested.